# Long Non-Coding RNA and microRNA Interplay in Colorectal Cancer and Their Effect on the Tumor Microenvironment

**DOI:** 10.3390/cancers14215450

**Published:** 2022-11-05

**Authors:** Marie Rajtmajerová, Andriy Trailin, Václav Liška, Kari Hemminki, Filip Ambrozkiewicz

**Affiliations:** 1Laboratory of Translational Cancer Genomics, Biomedical Center, Faculty of Medicine in Pilsen, Charles University, Alej Svobody 1665/76, 323 00 Pilsen, Czech Republic; 2Laboratory of Cancer Treatment and Tissue Regeneration, Biomedical Center, Faculty of Medicine in Pilsen, Charles University, Alej Svobody 1665/76, 323 00 Pilsen, Czech Republic; 3Department of Surgery, University Hospital in Pilsen and Faculty of Medicine in Pilsen, Charles University, Alej Svobody 80, 323 00 Pilsen, Czech Republic; 4Department of Cancer Epidemiology, German Cancer Research Center, Im Neuenheimer Feld 280, 69120 Heidelberg, Germany

**Keywords:** lncRNA, miRNA, tumor microenvironment, colorectal cancer, CRC, lymphocytes, macrophages, CAFs

## Abstract

**Simple Summary:**

The interplay between the tumour and the immune system plays a vital role in disease progression. There are several mechanisms the tumour imposes to repress the immune response. Dysregulation of non-coding RNAs, the main role of which is to regulate transcription and translation, leads to expressional changes in various signaling molecules and can subsequently affect infiltration and/or activation of immune cells. Here, we focus on the miRNA/lncRNA/protein regulatory axes, which are known to influence the immune response in colorectal cancer.

**Abstract:**

As the current staging and grading systems are not sufficient to stratify patients for therapy and predict the outcome of the disease, there is an urgent need to understand cancer in its complexity. The mutual relationship between tumour and immune or stromal cells leads to rapid evolution and subsequent genetic and epigenetic changes. Immunoscore has been introduced as a diagnostic tool for colorectal cancer (CRC) only recently, emphasising the role of the specific tumor microenvironment in patient’s prognosis and overall outcome. Despite the fact that non-coding RNAs (ncRNAs), such as microRNAs (miRNAs) and long non-coding RNAs (lncRNAs), cannot be translated into proteins, they significantly affect cell’s transcriptome and translatome. miRNA binding to mRNA efficiently blocks its translation and leads to mRNA destruction. On the other hand, miRNAs can be bound by lncRNAs or circular RNAs (circRNAs), which prevents them from interfering with translation. In this way, ncRNAs create a multi-step network that regulates the cell’s translatome. ncRNAs are also shed by the cell as exogenous RNAs and they are also found in exosomes, suggesting their role in intercellular communication. Hence, these mechanisms affect the tumor microenvironment as much as protein signal molecules. In this review, we provide an insight into the current knowledge of the microenvironment, lncRNAs’, and miRNAs’ interplay. Understanding mechanisms that underlie the evolution of a tissue as complex as a tumour is crucial for the future success in therapy.

## 1. Introduction

In 2020, colorectal cancer (CRC) was the third most prevalent cancer worldwide, accounting for 10% of all cancer cases, and the second deadliest one [1]. CRC incidence varies greatly among different geographical regions and is generally higher in the transitioned countries. This disproportion is caused by lifestyle changes which come together with socioeconomic growth, especially concerning dietary factors and physical activity [1].

Current cancer therapies are based on the presumption that mutational changes of tumour-suppressant genes and oncogenes are responsible for the development of cancer. However, epigenetic factors have to be taken into consideration as they significantly affect the final translatome of a cell and so its fitness.

miRNAs are short (about 18–25 nucleotides), single-stranded, and stable RNAs which regulate gene expression at the post-transcriptional level [2]. They bind 3’UTR of their target gene mRNA and subsequently inhibit its translation or lead to mRNA degradation [2]. Specific miRNA signatures have been identified in various cancers which distinguish them from healthy tissue, thus, miRNA dysregulation affects cellular translatome. However, certain miRNAs can have tumour-suppressive effect in some cancers but oncogenic in others, hence their function depends on the environment [2]. miRNAs are produced from longer precursor molecules which undergo a maturation process starting in the nucleus and completed by the actions of Dicer and Argonaut proteins in the cytoplasm (Figure 1).

In contrast, long non-coding RNAs (lncRNAs) are more than 200 nucleotides long. They are involved in transcriptional as well as post-transcriptional regulation. Due to the fact that many lncRNAs can bind, or “sponge”, miRNAs to prevent them from binding to their target molecules, these two subgroups should always be investigated side by side, especially when it comes to functional studies. In fact, there is a plethora of competing endogenous RNAs (ceRNAs) including lncRNAs, circular RNAs (circRNAs), mRNAs, and transcribed pseudogenes, which all compete for the same set of miRNAs produced in a particular cell [3]. A recent study of RNA-Seq, miR-Seq, and clinical data from the TCGA identified 205 lncRNAs and 345 miRNAs differentially expressed in the colon adenocarcinoma. Out of these, 14 lncRNAs, and 39 miRNAs affected patient’s overall survival [4].

To fully understand the complexity of the tumour tissue, it is also important to identify which cell types are producing particular non-coding RNAs (ncRNAs) as these can have different effects in different contexts. The tumour is composed of many cell types, including the malignant cells, cancer-associated fibroblasts, and many types of immune cells such as T cells, B cells, macrophages, and NK cells. All of these cells create a specific tumour microenvironment (TME), which plays an important role in the tumourigenesis and significantly affects the disease outcome. Differences in TME may also provide explanation of different therapy outcomes in otherwise clinically comparable patients.

Several lncRNA-miRNA-protein axes have been proven to significantly affect TME. They can do so in many different ways, one of them being dysregulation of immune checkpoint expression. For example, PD1-PD-L1 immunosuppressive signalling, which prevents T cell activation, is regulated by several ncRNAs such as MIR22HG [5] or miR-156-5p [6]. Similarly, dysregulation of MHC I expression by miR-148a-3p [7] and miR27a [8] is another example of ncRNAs playing a crucial role in immune cell activation or repression via dysregulation of their inhibitory molecules on the surface of tumour cells. However, ncRNAs regulate not only activation of immune cells, but also affect the infiltration of these cells into the tumour as they for example interfere with important signalling pathways such as Wnt/β-catenin [9] or MYC [10] signalling. Understanding these regulatory mechanisms is vital for future success of anti-cancer therapies, as ncRNAs provide us with a great potential of not only therapeutic targets, but also diagnostic, prognostic, and stratification factors.

## 2. Key Signalling Pathways in CRC Carcinogenesis

The fine balance between pro- and anti-growth and pro- and anti-apoptotic signals is crucial to maintain homeostasis. Several signalling pathways are involved in this process and their components are often mutated in cancer leading to their dysregulation, but ncRNAs up- or downregulation affects these signalling pathways equally.

### 2.1. Wnt/β-Catenin Signalling

Amongst the highly evolutionarily conserved pathways, Wnt signalling dysregulation is often found in CRC, as well as in many other cancers [11]. The Wnt pathway is responsible for cells’ growth and proliferation during embryogenesis as well as in adult tissues [11]. Wnt signalling is either canonical or noncanonical, depending on the involvement of β-catenin. When released from the membrane-bound complex, β-catenin is either degraded in proteosome, or when Wnt signalling is active it is transported into the nucleus where it serves as a transcription factor [12]. β-catenin nuclear accumulation is associated with more aggressive disease [12]. In a teratoma model, Wnt activation also negatively correlates with T cell and B cell infiltration, which is otherwise associated with worse clinical outcome [9].

Dysregulation of several ncRNAs in CRC is associated with Wnt/β-catenin upregulation, leading to subsequent clinical impact. Namely, lncRNA CCAL is a key player in the CRC tumourigenesis. Its high expression correlates with worse overall survival and response to therapy. This is due to CCAL’s ability to downregulate AP-2α which activates the Wnt/β-catenin pathway [13]. This signalling pathway is also triggered by another lncRNA, HCG18, which is upregulated in CRC patient samples as well as in CRC cell lines and its expression is inversely correlated to miR-1271 expression. HCG18 can sponge miR-1271 and its inhibition leads to restrained tumour growth and invasion [14]. The Wnt/β-catenin pathway is also activated by TDRKH-AS1 upregulation in CRC [15]. LncRNA H19, which is commonly upregulated in CRC, targets miR-29b-3p. By sponging miR-29b-3p, H19 upregulates its target protein PGRN, which in turn upregulates the downstream Wnt signalling to promote epithelial-mesenchymal transition (EMT) [16].

### 2.2. MYC Signlling

MYC is a transcription factor whose activity is regulated by several signalling pathways including the Wnt pathway, MAPK signalling activated by growth and survival factors, Jak/STAT signalling, and the TGF-β signalling pathway [10]. MYC is upregulated by various mechanisms (mutation, amplification, protein stabilisation, etc.) in more than 70 % of human cancers, leading to persistent proliferative as well as antiapoptotic signalling [10]. In a liver cancer model, MYC upregulation was shown to induce innate cell infiltration as well as enhanced checkpoint inhibitor expression by tumour cells, which altogether led to increased angiogenesis, chemoresistance, and proliferation of tumour cells [10].

In CRC, c-MYC activation was induced by lncRNA lncCMPK2. lncCMPK2 is localised in the nucleus and binds FUBP3 (far upstream element binding protein 3). In such a complex it guides FUBP3 to the FUSE (far upstream element) of c-Myc, whose transcription is subsequently activated. lncCMPK2 is upregulated in CRC cell lines compared to normal epithelium and its upregulation in patient samples correlates with tumour size, regional lymph node metastases, and TNM staging [17].

LncRNA GLCC1 is also known to accelerate CRC carcinogenesis as it stabilises c-Myc and prevents it from ubiquitination [18].

On the other side, MYC targets’ downregulation and TNF-α/NFκB, TGF-β, IFN-α, and IFN-γ signalling gene signatures and the subsequent insufficient differentiation is caused by high MIR31HG (lncRNA precursor molecule of miR31-3p and miR31-5p) expression [19]. Moreover, MIR31HG is a prognostic marker and a potential stratification factor for CRC, which is independent of tumour infiltration by cytotoxic T-lymphocytes or fibroblasts [19]. miR31-5p itself is a well-known poor prognostic factor in CRC [20] and miR-31-3p induces cetuximab resistance in metastatic CRC [21]. 

MYC binding protein (MYCBP) is another c-Myc-associated protooncogene. It is negatively regulated by miR-495-3p. In CRC, upregulation of lncRNA LUNAR1 reactivates MYCBP by miR-495-3p sponging. LUNAR1 knockdown leads to restriction of proliferation, migration, and progression in CRC cells, and promotes apoptosis [22].

### 2.3. MAPK Signalling

lncRNA SLCO4A1-AS1 is also upregulated in CRC and its high levels are associated with poor prognosis. It was also found to promote proliferation, migration, and invasion through activation of the epidermal growth factor receptor (EGFR)/mitogen-activated protein kinase (MAPK) signalling pathway [23]. MAPK8 is also activated by the sponging effect of PVT1 (plasmacytoma variant translocation 1) lncRNA on miR-152-3p, whose target E2F3 passes the signal directly to MAPK8 [24]. 

### 2.4. Shh Signalling

The hedgehog signalling pathway is another evolutionarily conserved pathway which is crucial for the maintenance of stem-like properties and proper differentiation. The Sonic hedgehog (Shh) protein disrupts the inhibition of a G protein-coupled receptor called Smo. Its action then leads to Gli1 transcription factor translocation into the nucleus, which starts the transcription of various target genes—amongst others, the Gli1 gene, to introduce a positive feedback loop [25]. Hedgehog signalling crosstalks with other important pathways involved in proliferation and differentiation, especially Wnt, EGFR, and Notch pathways [25].

In CRC, hedgehog signalling is activated by lncRNA-cCSC1. It is highly expressed in colorectal cancer stem cells. lncRNA-cCSC1 depletion leads to decreased self-renewing potential of CRC stem cells and to increased sensitivity to 5-fluorouracil [26], which is commonly used for CRC therapy. lncRNA-cCSC1 is thus a potential therapeutic target. 

ncRNAs involved in dysregulation of crucial signalling pathways are summarizes in Table 1.

## 3. Chemoresistance-Associated ncRNAs

The treatment strategy in a particular patient is dependent mainly on the stage and grade at which the disease is firstly diagnosed. For stage I, surgical or endoscopic resection is usually sufficient, whereas in more developed tumours chemotherapy has to be applied. 5-fluorouracil is the first-choice treatment; others include irinotecan, oxaliplatin, or capecitabine. Most commonly used combination agents are FOLFOX (5-fluorouracil + leucovorin + oxaliplatin), FOLFIRI (folic acid + 5-flurorouracil + leucovorin + irinotecan), and FOLFOXIRI (folic acid + 5-fluorouracil + leucovorin + oxaliplatin + irinotecan) [29]. Despite ongoing research in this field, chemoresistance arising in cancer cells under evolutionary pressure is still a crucial obstacle in therapy. Together with precise patient stratification, overcoming the chemoresistance is the key to therapy success.

Recent studies prove that specific ncRNAs may be the key targets in overcoming chemotherapeutic resistance. In case of the lncRNA subgroup in CRC these are for example HOTAIRM1 [30], lncRNA-cCSC1 [26], SCARNA2 [31], LINC01347 [32], HAND2-AS1 [33], lnc273–31, and lnc273–34 [34], TUG1 [35].

Several ncRNAs have been shown to induce resistance to 5-fluorouracil in CRC, namely lncRNA-cCSC1 [26], as mentioned previously. Amongst others, lncRNA SCARNA2, which is upregulated in CRC, was proven to induce 5-fluorouracil resistance via the miR-342-3p-EGFR/BCL2 axis. Its downregulation in vitro significantly enhanced CRC cell line sensitivity to this drug [31]. A similar effect was observed for LINC01347, which is responsible for LOXL2 upregulation in CRC via miR-328-5p attenuation [32]. 

Compared to normal tissue, lncRNA HOTAIRM1 expression is lower in CRC tissue samples and cell lines, and it is actually even more repressed in the cells resistant to 5-fluorouracil [30]. In vitro, multi-drug resistance, but also viability, migration, and proliferation of CRC cells were reduced by an increase in HOTAIRM1 expression. HOTAIRM1 targets miR-17-5p itself has opposite effects on CRC cells [30]. miR-17-5p is commonly upregulated in CRC, and enhances EMT and metastatic potential [36].

Similarly, lncRNA heart and neural crest derivatives-expressed 2-antisense RNA 1 (HAND2-AS1) is downregulated in 5-fluorouracil resistant CRC cells. In those which are sensitive, it sponges miR-20a in order to upregulate its protein target programmed cell death factor 4 (PDCD4), which leads to tumour growth inhibition [33].

5-fluorouracil resistance can also be induced by hypoxic conditions within the tumour. The presumption is that hypoxia, as a stress factor, activates a specific molecular response which subsequently induces resistance. This was proven for example for miR-675-5p, induced by hypoxic conditions, which disrupts the apoptotic signals induced by 5-fluorouracil targeting pro-caspase-3 [37].

Oxaliplatin resistance was investigated in an extensive study conducted by Song et al. [34]. Together with another 35 upregulated and 4 downregulated differentially expressed lncRNAs, lnc273–31 and lnc273–34 were induced by R273H mutation in the TP53 gene. They are responsible for oxaliplatin resistance (the levels of lnc273–31 even increased in a dose-dependent manner after exposure) and when deleted, tumour initiation, invasion, EMT, and chemoresistance decrease significantly. lnc273–31 and lnc273–34 are also crucial for stem-like properties’ maintenance [34]. TUG1 is another lncRNA inducing oxaliplatin resistance in CRC stem cells, where its expression is elevated. It is proven to stabilise transcription factor GATA6 [35]. 

On the other side, miR-27b-3p was proven to sensitise CRC cells to oxaliplatin treatment by inhibiting autophagy by downregulating ATG10. miR-27b-3p itself is inhibited by c-Myc at the transcriptional level [38].

## 4. Communication among Various Cell Types within the Tumour

### 4.1. Communication between Cancer-Associated Fibroblasts and Tumour

Cancer-associated fibroblasts (CAFs) are an important cellular subset of the TME as they produce various signal molecules and affect cancer cells. In the case of ncRNAs they usually do so by releasing them in exosomes which later fuse with the target cell. In CRC, such exosomes were found to promote tumour progression, metastatic spreading, and chemoresistance [27,28,39]. However, the relationship is mutual. For example, cancer cell-derived exosomes may contain miRNAs which in turn stimulate CAFs to promote metastasis, which was proven in case of miR-146-5p and miR-155-5p [40].

The aforementioned lncRNA CCAL, which promotes oxaliplatin resistance of CRC cells [13], is an example of ncRNA carried in CAFs-derived exosomes. Inside the CRC cells it acts as a HuR (human antigen R) mRNA stabiliser, which effectively increases β-catenin levels [27]. 

Similar results were obtained for H19. In the mouse model as well as in CRC patient samples, H19 was significantly upregulated in the tumour tissue compared to the adjacent normal colon. According to RNA-FISH experiments, H19 expression was localised in tumour stroma rather than the tumour itself [28]. Altogether, H19 is a key player in the tumourigenesis as it induces CRC stemness and chemoresistance in vitro as well as in vivo. It also activates the β-catenin pathway via sponging miR-141 [28], which otherwise inhibits CRC stemness. Interestingly, H19 is regulated by the proviral integration site for Moloney murine leukaemia virus (PIM) kinases by H19 promoter methylation. H19 expression subsequently leads to SOX2, OCT-4, and NANOG expression, as verified in the T cell acute lymphoblastic leukaemia microarray study [41].

miR-93-5p was found to be enriched in CAF’s derived exosomes compared to those isolated from adjacent tissue. miR-93-5p prevents cancer cell apoptosis after irradiation as it downregulates FOXA1, leading to TGB3 upregulation [42]. 

miR-24-3p is responsible for chemotherapy resistance in various tumours. In CRC it is preferentially expressed in CAFs’-derived exosomes, leading to CDX2 and HEPH downregulation. Its inhibition greatly decreases CRC cells’ viability and so it prevents the tumour growth [43].

CAF-derived exosomes were also found to be enriched in miR-17-5p. miR-17-5p introduces a positive feedback signalling loop via RUNX3/MYC/TGF-β1 axis to activate CAFs, which is associated with greater metastatic potential and poor overall survival [44]. 

### 4.2. Communication between Lymphocytes and Tumour

The type of immune response elicited by the tumour can significantly reduce or enhance tumour growth and spread. In fact, an immunoscore was established in 2018 for CRC as a reliable prognostic tool [45], even more accurate than the traditional TNM staging system. It consists of the quantification of CD3+ and CD8+ T-lymphocytes in the tumour centre and its advancing margin area [45]. 

Mechanistically, infiltration is only enabled when a sufficient capillary network is present. LINC01116 expression was proven to inactivate TPM1 promoter through EZH2 expression, which subsequently promotes CRC cells’ proliferation and angiogenesis [32].

There are several mechanisms tumour cells impose to repress the immune response, which is crucial for tumour elimination. Amongst others, ncRNAs play role in this immunosuppression. For example, CD8+ T-lymphocytes are suppressed by lncRNA HCG18 (also mentioned previously) via PD-L1 by sponging its inhibitor miR-20b-5p [46]. HCG18 also promotes cetuximab resistance in CRC cells [46]. Its target miR-20b-5p is also involved in other axes in the CRC progression. It is also outcompeted by MALAT1 to maintain stem-like properties of CRC cells by Oct4 activation [47]. miR-20b-5p also targets HIF1A (hypoxia-inducible factor 1 alpha) and when sponged by lncRNA COL4A2-AS1, this axis promotes proliferation and glycolysis in CRC cells [48].

MIR22HG is significantly downregulated in CRC, which correlates with worse overall survival and decreased T-cell counts within the tumour [5]. Its tumour-suppressive function is mediated by outcompeting SMAD2, which disrupts the TGFβ signalling pathway. Contrary, MIR22HG upregulation correlates with CD8A expression and also CD8+ T cell infiltration [5]. PD-L1 expression was also proven to be downregulated by blocking IL-17. This blockade leads apart from others to miR-15b-5p expression which sequesters PD-L1, leading to improved T cell infiltration [6].

MHC I downregulation is a crucial way for tumour cells to escape CD8+ T cell-mediated immune response. Contrary, normal expression of MHCI ensures proper immune function. This can be restored by miR-148a-3p inhibition. When blocked, miR-148a-3p is not able to bind calnexin (CANX), which helps to fold MHC I protein [7]. CD8+ T cell activation is also blocked by IDO1 expression in CRC. IDO1 is in turn repressed by miR-448, leading to tumour suppression [49]. Similarly, miR-27a is proven to repress calreticulin, which is another chaperone of MHC I molecules [8]. Tumours with high expression of miR-27a show low expression of MHC I and CD8+ T cell infiltration and poor prognosis [8].

A comprehensive bioinformatic study of 568 CRC patient samples from the TCGA database revealed that low MIR4435-HG expression is associated with higher plasma cells and CD4+ resting memory T cell infiltration. On the contrary, high MIR4435-HG expression is associated with neutrophilic and follicular helper T cell infiltration [50]. ELFN1-AS1^high^ group showed higher infiltration of plasma cells, CD4+ memory activated T cells, gamma-delta T cells, monocytes, and myeloid dendritic cells. Conversely, ELFN1-AS1^low^ patient samples were highly infiltrated by neutrophils and CD8+ T cells [50]. 

lnRNA KCNQ1OT1 is upregulated in CRC in correlation with CD155, which is known to have immune-suppressive function in CRC [51]. Despite high CD8+ T cell infiltration, lncRNA KCNQ1OT1 and CD155 high expression are associated with poor prognosis [51].

In fact, lncRNA KCNQ1OT1 secreted in tumour-derived exosomes is able to regulate PD-L1 expression and induce immune evasion [52]. By sponging miR-30a-5p it regulates PD-L1 ubiquitination (through USP22 upregulation) to inhibit CD8+ T cell function [52]. 

A recent study based on publicly available data showed that LINC00657 negatively correlates with CD8+ T cell infiltration in CRC [53]. These results were verified in vivo and a competitive endogenous RNA network was investigated, comprising of hsa-miRNA-1224-3p and hsa-miRNA-338-5p and SCD, ETS2, UBE2H, and YY1 [53].

However, our understanding of the mutual relationship between cancer cells and lymphocytes and the ncRNA signalling network is not sufficiently comprehensive so far and thus remains to be clarified in more depths. Many lncRNA-protein-TME and miRNA-protein-TME interactions have been discovered, but only few lncRNA-miRNA-protein-TME axes are known. 

### 4.3. Communication between Macrophages and Tumour

Tumour-associated macrophages (TAMs) are the most abundant type of immune cells present in CRC TME. They secrete exosomes and cytokines in order to enhance (M1 macrophages) or suppress (M2 macrophages) the anti-tumour immune response (by recruiting Treg cells) and promote angiogenesis, tumour growth, and invasion [54]. Despite their significance, very few studies have been conducted to investigate the lncRNA-miRNA-TAMs network in CRC.

In 2019, Qu et al. proved that miR-133a-3p, which targets RhoA, was significantly decreased in CXCR4^high^ HCT116 CRC cells. miR-133a-3p is sponged by lncRNA XIST and XIST is in turn upregulated by CXCL12/CXCR4 axis [55]. CXCR4 overexpression was proven to drive myeloid-derived suppressor cells (MDSCs) and macrophage infiltration in the colonic tissue, which favours tumourigenesis by suppressing the immune response [55]. Moreover, CXCR4 signalling triggers a cascade leading to anti-apoptotic Bcl-2 signalling and surviving production [56]. Hence, it increases tumour cell resistance. Interestingly, high CXCR4 expression was observed in ATPase inhibitory factor 1 (IF1) silenced cells. IF1 overexpression was also associated with higher NK cell infiltration [57].

In another study, lncRNA MIR155HG was shown to drive macrophage polarisation towards the immunosuppressive M2 phenotype. Its high expression in CRC cells is correlated with Annexin A2 (ANXA2) overexpression. That is explained by the fact that MIR155HG is a molecular sponge of miR-650, which in turn targets ANXA2. Moreover, both MIR155HG and ANXA2 knockdown led to reduced M2 macrophage polarisation and CRC progression in nude mice [58]. The M2 phenotype is also induced by miR-21-5p and miR-200a excreted by CRC cells in small extracellular vesicles [59]. They are also responsible for increased PD-L1 expression and the consecutive T cell response attenuation [59].

Contrary, miR-195-5p was proven to prevent M2 polarisation by downregulating IL-4 production via its interaction with NOTCH2 [60]. miR-195-5p downregulation in CRC tissue was associated with worse overall survival [60].

## 5. Conclusions

As the incidence of CRC is rising worldwide, there is an urgent need to understand the mechanisms driving CRC tumourigenesis in order to target them in therapy. Whilst the driver mutations of protein coding genes are well described, ncRNAs have only recently begun to gain more attention. Their ability to modulate TME in CRC is indisputable and various lncRNA-miRNA-TME pathways have already been depicted (Table 2). However, as we have mentioned previously, many of these pathways are not completely understood and only specific miRNA-TME or lncRNA-TME associations are known. More complex understanding of regulatory processes participating in CRC will be obtained once these gaps are discovered. This is especially important given the fact that several ncRNAs involved have therapeutic potential. Many ncRNAs are also notably important for their prognostic and diagnostic value.

The overall understanding of ncRNA dysregulation driving TME and the way these changes contribute to tumour progression remains to be discovered, especially when we consider that up to this day, no studies have focused on the role of any other cell type in CRC TME apart from lymphocytes, TAMs, or CAFs.

## Figures and Tables

**Figure 1 cancers-14-05450-f001:**
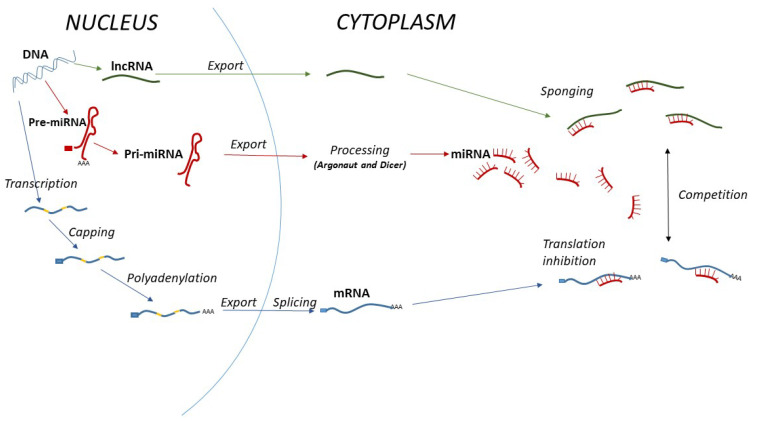
lncRNA–miRNA interaction. Similarly to mRNAs, miRNAs undergo a series of processing steps in the nucleus as well as in the cytoplasm. Some miRNAs can also be produced from their lncRNA precursors. Fully matured mRNAs as well as all non-coding RNAs (such as miRNAs and lncRNAs) are localised in the cytoplasm where lncRNAs and mRNAs compete for miRNAs. This binding can either block or trigger the final translational process and is therefore an important post-transcriptional regulatory step in gene expression.

**Table 1 cancers-14-05450-t001:** ncRNA-related regulation of key signalling pathways.

Signalling Pathway	lncRNA	miRNA	Protein
Wnt/β-catenin	CCAL		HuR [27], AP-2α [13]
	H19	miR-141	β-catenin [28]
		miR-29b-3p	PGRN [16]
	HCG18	miR-1271 [14]	Not specified
	TDRKH-AS1 [15]	Not specified	Not specified
MYC	lncCMPK2 [17]	Not specified	Not specified
	GLCC1 [18]	Not specified	Not specified
	LUNAR1	miR-495-3p [22]	Not specified
MAPK	SLCO4A1-AS1 [23]	Not specified	Not specified
	PVT1	miR-152-3p [24]	E2F3
SHH	lncRNA-cCSC1 [26]	Not specified	Not specified

Green: ncRNA has a positive (activatory) effect on that particular signalling pathway Red: ncRNA has a negative (inhibitory) effect on that particular signalling pathway. Not specified: Data is either missing or was not included in the experimental design at all.

**Table 2 cancers-14-05450-t002:** The most important lncRNA–miRNA axes impacting TME and drug resistance in CRC.

LncRNA	Produced by	Acting on		Effect on TME
		miRNA	Protein	
CCAL	CAFs	Not specified	HuR	β-catenin stabilisation [27]
	Not specified	Not specified	AP-2α	Wnt/β-catenin activation [13]
H19	CAFs	miR-141	β-catenin	Wnt/β-catenin activation [28]
		Not specified	SOX2, OCT-4, and NANOG	Induce stemness [41]
	Not specified	miR-29b-3p	PGRN	Wnt/β-catenin activation and EMT promotion [16]
LUNAR1	Not specified	miR-495-3p	MYCBP	Proliferation, migration, progression [22]
PVT1	Not specified	miR-152-3p	E2F3	MAPK signalling [24]
HCG18	Not specified	miR-20b-5p	PD-L1	CD8^+^ T-lymphocytes suppression, cetuximab resistance [46]
MALAT1	Not specified	miR-20b-5p	Oct-4	Induce stemness [47]
COL4A2-AS1	Not specified	miR-20b-5p	HIF1A	Proliferation, glycolysis [48]
HCG18	Not specified	miR-1271	Not specified	Wnt/β-catenin activation [14]
LINC01116	Not specified	Not specified	EZH2	TPM1 promoter inactivation, enhanced proliferation and angiogenesis [32]
XIST	Not specified	miR-133a-3p	RhoA	MDSCs and macrophages infiltration [55]
MIR155HG	Not specified	miR-650	ANXA2	Macrophage polarisation towards M2 [58]
SCARNA2	Not specified	miR-342-3p	EGFR/BCL2	5-fluorouracil resistance [31]
LINC01347	Not specified	miR-328-5p	LOXL2	5-fluorouracil resistance [61]
HAND2-AS1	Not specified	miR-20a	PDCD4	5-fluorouracil sensitivity [33]
HOTAIRM1	Not specified	miR-17-5p	BTG3	5-fluorouracil sensitivity [30]
TUG1	Not specified	Not specified	GATA6	Oxaliplatin resistance [35]
MIR22HG	Not specified	Not specified	PDL1, CD8A , SMAD2	CD8+ T cell infiltration, TGFb signalling disruption [5]
Not specified	CAFs	miR-93-5p	FOXA1, TGB3	Radioresistance [42]
Not specified	CAFs	miR-24-3p	CDX2, HEPH	Chemoresistance [43]
Not specified	CAFs	miR-17-5p	RUNX3	CAF activation via TGF-β1 upregulation, MYC upregulation [44]
Not specified	Not specified	miR-15b-5p	PD-L1	Higher T cell infiltration [6]
Not specified	Not specified	miR-148a-3p	CANX	Incorrect MHCI folding and subsequent MHCI downregulation [7]
Not specified	Not specified	miR-448	IDO1	CD8+ T cell activation blockade [49]
Not specified	Not specified	miR-27a	CALR	Incorrect MHCI folding and subsequent MHCI downregulation [8], poor prognosis
KCNQ1OT1	Not specified	miR-30a-5p	USP22	Upregulation leads to PD-L1 dysregulation, immune evasion, and poor prognosis [51,52]
LINC00657	Not specified	miRNA-1224-3p, miRNA-338-5p	SCD, ETS2, UBE2H, YY1	Lower CD8+ T cell infiltration [53]
Not specified	CRC cells	miR-21-5p, miR-200a		M2 phenotype induction, increased PD-L1 expression [59]
Not specified	Not specified	miR-195-5p	NOTCH2	IL-4 downregulation leading to M2 phenotype inhibition [60]
MIR4435-HG ^low^	Not specified	Not specified	Not specified	Plasma B cells and CD4+ resting memory T cell infiltration [50]
MIR4435-HG ^high^	Not specified	Not specified	Not specified	Neutrophils and follicular helper T cell infiltration [50]
ELFN1-AS1 ^high^	Not specified	Not specified	Not specified	Plasma B cells, CD4+ memory activated T cells, gamma-delta T cells, monocytes, and myeloid dendritic cells
ELFN1-AS1 ^low^	Not specified	Not specified	Not specified	Neutrophils and CD8+ T cells [50]

Green: lncRNA has a positive effect on that particular miRNA or protein, i.e., upregulation, stabilization etc. Red: lncRNA has a negative effect on that particular miRNA or protein, i.e., downregulation or sponging. Not specified: Data is either missing or was not included in the experimental design at all.

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
