# Peer review of "Long Non-Coding RNA and microRNA Interplay in Colorectal Cancer and Their Effect on the Tumor Microenvironment"

_cancers, 2022, doi:10.3390/cancers14215450_

Round 1

Reviewer 1 Report

In the review titled "Long non-coding RNA and microRNA interplay in colorectal cancer and their effect on tumor microenvironment," the authors summarized the ncRNAs affecting key signaling pathways, chemoresistance, and microenvironment in CRC carcinogenesis, providing an overall understanding of ncRNAs in CRC tumor progression.

Some minor suggestions

Line 57 :protooncogenic should be oncogenic. Proto-oncogenes are the normal genes that regulate cell division.

It might be better to separate lncRNA-miRNA affecting drug resistance from Table 2, since chemoresistance-associated ncRNAs were summarized in an independent section.

Author Response

Dear Reviewer,

We have prepared response to your comments. Please see the attachment.

Regards

Reviewer 2 Report

The authors focused on the lncRNA, miRNA, immune microenvironment, and formulated the interaction between long non-coding RNA and microRNA in colorectal cancer and their effect on tumor microenvironment systematically. It's a good work, and has certain value for subsequent research in this field. However, it is hoped that the author can increase the number of references appropriately, especially the part of “Communication among various cell types within the tumour”. This will make the article more convincing.

Author Response

(The authors gave the same response as above.)

Reviewer 3 Report

The review manuscript, Long non-coding RNA and microRNA interplay in colorectal cancer and their effect on tumor microenvironment, by Mr./Mrs. Marie Rajtmajerova et al. collects the previous studies about the effect of lncRNAs and miRNA in CRC and tried to organize them in this drift. Many studies indicated that disorder of lncRNA and miRNA  cause many diseases such as cancer. Several suggestions listed here hope to be advantage to this manuscript.

Comments

1.     The introduction of the interaction among lncRNA, miRNA and CRC in the text is too short to understand the importance in CRC progression. For example, several critical lncRNA regulatory pathways in CRC such as PD1/PD-L1 need to be introduced in detail and to do some discussion.

2.     About the effect of lncRNA and miRNA on tumor microenvironment is not very clear in this manuscript. Only show the Table and introduce it briefly. So many information on the tables are unknow and no discussion in the text. For example, the authors can use the bioinformatics tools to analyze the target miRNAs of the lncRNAs and do some discussion in the text.

3.     The discussion about the future work in this field needs to be discussion based on the studies.

4.     Overall, whole drift needs to be reorganized to improve this review manuscript.

Author Response

(The authors gave the same response as above.)
